# Relationship between Standard Uptake Values of Positron Emission Tomography/Computed Tomography and Salivary Metabolites in Oral Cancer: A Pilot Study

**DOI:** 10.3390/jcm9123958

**Published:** 2020-12-07

**Authors:** Shigeo Ishikawa, Toshitada Hiraka, Kazukuni Kirii, Masahiro Sugimoto, Hiroaki Shimamoto, Ayako Sugano, Kenichiro Kitabatake, Yuuki Toyoguchi, Masafumi Kanoto, Kenji Nemoto, Tomoyoshi Soga, Masaru Tomita, Mitsuyoshi Iino

**Affiliations:** 1Department of Dentistry, Oral and Maxillofacial Plastic and Reconstructive Surgery, Faculty of Medicine, Yamagata University, 2-2-2 Iida-Nishi, Yamagata 990-9585, Japan; shigeo_ishikawa2011@yahoo.co.jp (S.I.); a.sugano@yamatoku-hp.jp (A.S.); kenichiro5210@yahoo.co.jp (K.K.); m-iino@med.id.yamagata-u.ac.jp (M.I.); 2Department of Radiology, Division of Diagnostic Radiology, Faculty of Medicine, Yamagata University, 2-2-2 Iida-Nishi, Yamagata 990-9585, Japan; hirakatoshitada@gmail.com (T.H.); kazukunikirii@gmail.com (K.K.); c-elegans_0201g@mail.goo.ne.jp (Y.T.); mkanoto@med.id.yamagata-u.ac.jp (M.K.); 3Institute for Advanced Biosciences, Keio University, 246-2 Mizukami, Kakuganji, Tsuruoka, Yamagata 997-0052, Japan; soga@sfc.keio.ac.jp (T.S.); mt@sfc.keio.ac.jp (M.T.); 4Health Promotion and Preemptive Medicine, Research and Development Center for Minimally Invasive Therapies, Tokyo Medical University, Shinjuku, Tokyo 160-8402, Japan; 5Department of Oral and Maxillofacial Radiology, Osaka University Graduate School of Dentistry, 1-8 Yamadaoka, Suita 565-0871, Japan; h-shima@dent.osaka-u.ac.jp; 6Department of Radiology, Division of Radiation Oncology, Faculty of Medicine, Yamagata University, 2-2-2 Iida-Nishi, Yamagata 990-9585, Japan; knemoto@ymail.plala.or.jp

**Keywords:** saliva, metabolomics, ^18^F FDG PET, CT, biomarker

## Abstract

^18^F-fluorodeoxyglucose (^18^F-FDG) positron emission tomography (PET)/computed tomography (CT) is usually used for staging or evaluation of treatment response rather than for cancer screening. However, ^18^F-FDG PET/CT has also been used in Japan for cancer screening in people with no cancer symptoms, and accumulating evidence supports this application of ^18^F-FDG PET/CT. Previously, we have observed a correlation between the saliva and tumor metabolomic profiles in patients with oral cancer. Hence, if salivary metabolites demonstrate a significant correlation with PET parameters such as the maximum standardized uptake value (SUV_max_), they may have the potential to be used as a screening tool before PET/CT to identify patients with high SUV_max_. Hence, in this study, we aimed to explore the relationship between salivary metabolites and SUV_max_ of ^18^F-FDG PET/CT using previously collected data. ^18^F-FDG PET/CT was performed for staging 26 patients with oral cancer. The collected data were integrated and analyzed along with quantified salivary hydrophilic metabolites obtained from the same patients with oral cancer and controls (*n* = 44). In total, 11 metabolites showed significant correlations with SUV_max_ in the delayed phases. A multiple logistic regression model of the two metabolites showed the ability to discriminate between patients with oral cancer and controls, with an area under the receiver operating characteristic curve of 0.738 (*p* = 0.001). This study uniquely confirmed a relationship between salivary metabolites and SUV_max_ of PET/CT in patients with oral cancer; salivary metabolites were significantly correlated with SUV_max_. These salivary metabolites can be used as a screening tool before PET/CT to identify patients with high SUV_max_, i.e., to detect the presence of oral cancer.

## 1. Introduction

^18^F-fluorodeoxyglucose (^18^F-FDG) positron emission tomography (PET)/computed tomography (CT) is a valuable imaging technique for managing oral cancer [1]. ^18^F-FDG PET/CT is commonly used not only for staging but also for assessing the therapeutic effect and prognosis of oral cancer [1,2,3,4]. The maximum standardized uptake value (SUV_max_), generated during an integrated ^18^F-FDG-PET/CT scan, provides information on the metabolic activity of the tumor and is an index that is used to discriminate between malignant lesions and benign lesions. Generally, a high SUV_max_ indicates a high possibility of malignancy [5]. Recently, dual-phase ^18^F-FDG PET/CT has been used as an alternative method when other types of preoperative imaging cannot clearly distinguish between benign and malignant lesions [6,7,8,9]. Thus, both early and delayed SUV_max_ values are important for diagnosing malignancy.

The Warburg effect, a key metabolic mechanism exploited by ^18^F-FDG PET/CT [5,10], is a cancer-specific metabolic shift manifesting as increased glucose absorption by aerobic glycolysis activation [5,11]. We have previously revealed that salivary metabolites have the potential to discriminate between patients with oral cancer and healthy controls [12,13,14,15], indicating a possible correlation between salivary metabolites and SUV_max_. However, to the best of our knowledge, no studies have explored this relationship.

^18^F-FDG PET/CT is usually used for staging or evaluation of treatment response rather than for cancer screening. However, ^18^F-FDG PET/CT has also been used in Japan for cancer screening in people with no cancer symptoms, and accumulating evidence supports this application of ^18^F-FDG PET/CT [16,17,18,19]. Therefore, we believe that it is clinically meaningful to analyze the relationship between salivary metabolites and PET parameters such as SUV_max_ in patients with oral cancer. If salivary metabolites demonstrate a significant correlation with PET parameters such as SUV_max_, they may have the potential to be used as a screening tool before PET/CT to identify patients with high SUV_max_, i.e., the presence of oral cancer.

Hence, this study aimed to analyze the relationship between SUV_max_ and salivary metabolites in patients with oral cancer, which could reveal the utility and limitations of salivary metabolites as biomarkers.

## 2. Materials and Methods

### 2.1. Study Subjects

All patients were recruited from the Department of Dentistry, Oral and Maxillofacial Plastic and Reconstructive Surgery, Yamagata University Hospital. Patients enrolled in our previous studies [12,15] were included in this study. Briefly, none of the patients with oral cancer had received any treatment such as chemotherapy and radiotherapy prior to testing, had diabetes mellitus, or had a history of malignancy or an autoimmune disorder. All procedures performed in studies involving human participants were in accordance with the ethical standards of the institutional and/or national research committee and the Declaration of Helsinki 1964 and its later amendments or comparable ethical standards. The Ethics Committee of Yamagata University, Faculty of Medicine, approved this study protocol (approval number: 2012-141). Informed consent was obtained from all participants included in the study.

### 2.2. Metabolomics Data

The metabolomics data quantified in our previous study [12,15] were also used in this study. Briefly, saliva samples were collected between 8:00 a.m. and 12:00 p.m. Eating and drinking were not permitted for at least 1.5 h before the collection of saliva samples. Using oral hygiene products such as toothpaste and mouthwash was not permitted for at least 1.0 h before sample collection. Patients were asked to rinse their mouths with water immediately before saliva sample collection. On average, 400 μL of unstimulated whole saliva was collected. The metabolites in these saliva samples were quantified using capillary electrophoresis time-of-flight mass spectrometry (Agilent Technologies, Palo Alto, CA, USA).

### 2.3. FDG PET/CT Protocol

FDG PET/CT images were acquired according to a standard protocol—5 h of fasting before the injection of FDG (3.7 MBq/kg (body wight)), with blood glucose levels <150 mg/dL at the time of injection. At 60 min after FDG injection (early-phase images), whole-body PET/CT images were obtained under free-breathing conditions. A PET/CT system (Biograph mCT, Siemens Healthineers; Erlangen, Germany) that combined a full-ring PET scanner with lutetium oxyorthosilicate crystals and a 64-MDCT scanner was used for generating images. The same protocol was used 120 min after the injection of FDG for PET/CT images of the head and neck (delayed-phase images). CT studies were performed under breath-holding conditions with the following parameters: 120 kV; 80 mA; field of view: 780 mm; pitch: 1.2; and slice thickness: 0.6 mm. PET emission data were obtained in a three-dimensional (3D) mode with the following parameters: 2 min at each bed position (total 14 min); matrix size: 200 × 200; and Gaussian filter size: 5 mm. PET images were reconstructed using a 3D ordered subset expectation maximization algorithm (24 subsets and two iterations). All PET and CT images were integrated using an automatic image fusion software.

### 2.4. Collection of SUV_max_ Data

^18^F-FDG PET/CT was performed in all patients with oral cancer for staging, and dual-phase SUV_max_ was measured on dual-phase images by radiologists with expertise in this field. SUV_max_ values were confirmed by two expert radiologists and were calculated for the early and delayed phases. The SUV_max_ value for the delayed phase was not calculated for three patients.

### 2.5. Statistical Analyses

Spearman’s rank correlation coefficient was used to determine the correlations between SUV_max_ values and salivary metabolites. To determine the discrimination ability of multiple metabolite combinations, multiple logistic regression (MLR) analyses were performed using the backward elimination method. The prediction ability was evaluated using areas under receiver operating characteristic (ROC) curves (AUCs) to discriminate between patients with oral cancer and healthy controls. Only metabolites that were detected in >50% of the controls or patients with oral cancer were selected. Statistical significance was set at *p* < 0.05. Statistical analyses were performed using SPSS version 20 (IBM Corp., Armonk, NY, USA).

## 3. Results

Patient characteristics are shown in Table 1. The data on controls were the same as those reported in our previous studies [12,15]. Table 2 shows the salivary metabolites that were significantly and highly correlated with SUV_max_ (*p* < 0.05, Spearman’s rank correlation). Eleven salivary metabolites were correlated with delayed-phase SUV_max_^—^*N*-acetylneuraminate, pyruvate, hexanoate, homovanillate, 3-methylhistidine, 3-phenylpropionate, pipecolate, p-hydroxyphenylacetate, isethionate, crotonate, and o-phosphoserine. None of the metabolites showed a significant correlation with early-phase SUV_max_. Figure 1A shows the ROC curves for discriminating between patients with oral cancer and controls. The AUC was 0.738 (*p* = 0.001, 95% confidence interval 0.619–0.857). In order to evaluate the discrimination ability of this MLR model for the early stages (T1 and T2) and advanced stages (T3 and Tt), ROC curves were depicted at Figure 1B,C. Among the 11 metabolites, *N*-acetylneuraminate and 3-phenylpropionate were selected as independent variables in the MLR model using the backward elimination method. The *p*-values in the developed models were 0.018 and 0.018 for *N*-acetylneuraminate and 3-phenylpropionate, respectively. To evaluate the bias caused by sex, we developed another MLR model including *N*-acetylneuraminate, 3-phenylpropionate, and sex. The *p*-values for these features in this MLR model were 0.018, 0.017, and 0.344, respectively.

## 4. Discussion

This study aimed to determine the relationship between salivary metabolites and SUV_max_ of ^18^F-FDG PET/CT. ^18^F-FDG PET/CT is a cancer screening tool, but it has several limitations. Thus, there is a need for another screening tool that is complementary to ^18^F-FDG PET/CT. We have evaluated the screening potential of saliva samples and the correlations between SUV_max_ and salivary metabolites in patients with oral cancer. To our knowledge, this is the first report to reveal an association between SUV_max_ and salivary metabolites in patients with oral cancer. All salivary metabolites showed significant positive correlations with SUV_max_. Because high SUV_max_ values reflect increased anaerobic glycolysis metabolism in malignant tumors, the higher concentration of metabolites in patients with oral cancer in this study meant higher SUV_max_ values.

The salivary metabolites observed in this study may be used as a first-line screening tool to confirm the presence of oral cancer before PET/CT. MLR analyses with a backward elimination method identified that *N*-acetylneuraminate and 3-phenylpropionate had the highest AUCs for discriminating between patients with oral cancer and controls (AUC = 0.738, *p* = 0.001, Figure 1A). Therefore, these metabolites could be used for detecting the pathological accumulation of ^18^F-FDG using PET/CT and discriminating between patients with oral cancer and controls. These findings are consistent with those reported in our previous studies, which showed that these metabolites could be reasonable choices for oral cancer screening. To evaluate the discrimination ability for the early stages and the advanced stages, we depicted the ROC curves (Figure 1B,C). These curves also showed high AUC values 0.773 (*p* = 0.005) and 0.712 (*p* = 0.015). We also evaluated the effect of sex where the *p*-value of sex was 0.344 in the MLR model including these two metabolites and sex. Therefore, only these metabolites provided enough evidence to discriminate between patients with oral cancers and healthy controls.

We speculated that 11 candidate salivary metabolites could be used to detect the pathological accumulation of ^18^F-FDG in oral cancer tissue using PET/CT. Some of our candidate metabolites were related to the glycolytic pathway or were a part of it, suggesting a Warburg effect as a form of cancer metabolism. Pyruvate is a final product of glycolysis. Pipecolate is a product of lysine degradation. Lysine is a ketogenic amino acid that is metabolized to acetyl-coenzyme A in the tricarboxylic acid cycle. These metabolites have been reported as screening biomarkers for various cancers [14,20]. In addition, in our previous study [15], *N*-acetylneuraminate, 3-methylhistidine, and pipecolate were significantly increased in oral cancer tissue. Thus, it is reasonable to suggest that these salivary metabolites are potential biomarkers that could be used to detect the pathological accumulation of ^18^F-FDG in patients with oral cancer.

In this study, 11 salivary metabolites showed significant correlations with delayed-phase SUV_max_, but there was no significant correlation between any salivary metabolite and early-phase SUV_max_. Dual-phase ^18^F-FDG PET/CT has recently been used to distinguish between benign lesions and malignant lesions [6,21,22,23,24]. The SUV_max_ of malignant lesions increased in the delayed phase, whereas FDG uptake in most benign lesions decreased in the delayed phase [21]. Erdem et al. concluded that delayed-phase ^18^F-FDG imaging increased the detectability of the primary lesion because of higher FDG uptake by primary tumors in the delayed phase compared to that in the early phase of imaging [6]. Delayed-phase SUV_max_ has the potential to show the aggressiveness of the tumor better than early-phase SUV_max_. These facts may partly explain the lack of a significant correlation between any salivary metabolite and early-phase SUV_max_ in our study. In contrast, 11 salivary metabolites showed significant correlations with delayed-phase SUV_max_.

SUV_max_ is affected by several factors, including body weight, amount of time passed between the injection and the scanning, plasma glucose level, tumor size, and region of interest [25,26,27,28]. In this study, none of the patients were obese as their body mass index was 15.0–27.1 kg/m^2^; only three patients had a body mass index of >25.0 kg/m^2^. The time from FDG injection to scan was the same in all patients, and no patients had diabetes mellitus. Hence, the dispersion of the factors that could affect SUV_max_ values was minimal.

This study has several limitations. Firstly, SUV_max_ is occasionally high not only for malignant tumors but also for non-malignant lesions, such as lesions due to inflammation or benign tumors. In such cases, it may not be possible to distinguish a benign lesion from a malignant one based on SUV_max_ alone [5]. Hence, there are concerns regarding the low specificity of the metabolites, which would be expected to be enhanced in the presence of both oral cancer and inflammation, such as periodontal disease. Our previous study, however, revealed differences between oral cancer and periodontal disease [14], with periodontal disease having a lesser effect on the salivary metabolomics results than oral cancer [15]. Secondly, SUV_max_ determined by PET/CT is not always high for low glycolysis malignant tumors, such as well-differentiated lung adenocarcinoma, hepatocellular carcinoma, prostate cancer, kidney cancer, and gastric cancer [29,30,31]. It is unclear how salivary metabolites could change in the aforementioned lesions to produce a low SUV_max_. Thirdly, our candidate salivary metabolites may not be biomarkers specifically related to oral cancer. Generally, most cancers are associated with high SUV_max_.

Our previous salivary metabolomics studies have shown a large overlap of aberrant metabolites in oral, breast, and pancreatic cancer patients [14]. We also analyzed the consistently elevated metabolites in saliva and oral cancer tissue [15]. In the present study, we used a different approach to identify the metabolites showing the ability to discriminate between oral cancer and healthy controls based on the correlation between salivary metabolites and SUV_max_ of PET/CT. Evidence of the salivary metabolites associated with various cancers has recently been accumulated [32]; therefore we have to validate the specificity of these metabolites with the saliva samples collected from other cancers.

## 5. Conclusions

In summary, we confirmed the relationship between salivary metabolites and SUV_max_ of PET/CT in patients with oral cancer; the salivary metabolites significantly correlated with SUV_max_. These salivary metabolites can be used as a screening tool before PET/CT to identify patients with high SUVmax, i.e., the presence of oral cancer.

## Figures and Tables

**Figure 1 jcm-09-03958-f001:**
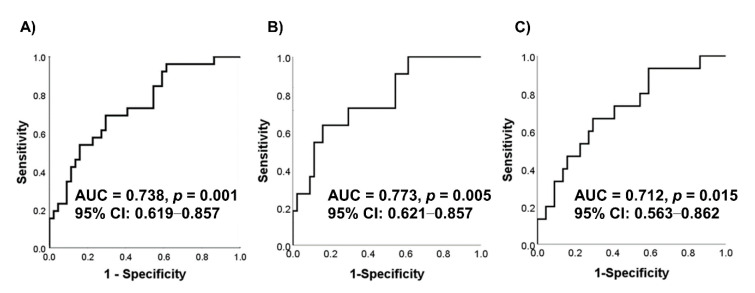
ROC curves of the MLR model to evaluate a combination of *N*-acetylneuraminate and 3-phenylpropionate to discriminate between patients with oral cancer (*n* = 22) and healthy controls. The ROC curve of the MLR model for discriminating the patients with oral cancer (*n* = 22) from healthy controls (*n* = 46) is depicted (**A**). Those curves for patients with oral cancer with T1 and T2 (*n* = 11) (**B**) and T3 and T4 (*n* = 15) (**C**) are depicted. ROC: the receiver operating characteristic curve; CI: confidence interval; MLR: multiple logistic regression; ROC: Receiver operating characteristic.

**Table 1 jcm-09-03958-t001:** Characteristics of subjects.

Parameter		Oral Cancer	Control
*n*		26	44
Age	Average (SD)	73 (29–94)	68 (21–90)
Sex	male	12 (46.2)	16 (36.4)
	female	14 (53.8)	28 (63.6)
Smoking habit	yes	14 (53.8)	9 (20.5) **
Periodontitis	yes	17 (65.4)	29 (65.9)
Stage	I	5 (19.2)	
	II	6 (23.1)	
	III	7 (26.9)	
	IV	8 (30.8)	
Standard Uptake Value	early phase (*n* = 26)	10.2 (2.8–19.4)	
	delayed phase (*n* = 23)	11.3 (1.8–22.4)	
Histological type	Squamous cell carcinoma	22 (84.6)	
	Malignant melanoma	2 (7.7)	
	Adenoid cystic carcinoma	1 (3.8)	
	Unknown	1 (3.8)	

Parentheses numbers were min–max for age and percentage of each group for the other parameters. SD, Standard deviation. ** *p* = 0.008.

**Table 2 jcm-09-03958-t002:** Pearson’s correlation analysis between salivary metabolites and SUV_max_ derived from PET/CT of oral cancer patients.

Salivary Metabolites	SUV_max_ in Delayed Phase	
	Spearman’s Rank Correlation Coefficient (*R*)	*p*-Value
*N*-acetylneuraminate	0.492	0.017
Pyruvate	0.467	0.025
Hexanoate	0.484	0.019
Homovanillate	0.455	0.029
3-methylhistidine	0.473	0.023
3-phenylpropionate	0.462	0.026
Pipecolate	0.521	0.011
p-hydroxyphenylacetate	0.453	0.030
Isethionate	0.482	0.020
Crotonate	0.417	0.048
*o*-phosphoserine	0.468	0.024

SUV_max_, maximum standard uptake value. *R* is a measure of the correlation coefficient.

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
