# Peer review of "Relationship between Standard Uptake Values of Positron Emission Tomography/Computed Tomography and Salivary Metabolites in Oral Cancer: A Pilot Study"

_jcm, 2020, doi:10.3390/jcm9123958_

Round 1

Reviewer 1 Report

This study rises the issue of screening (in patients with risk factor in oral cancer) before the execution of PET/CT. This could be very interesting because of the lack of specificity in FDG PET (tumour vs inflammation). For this reason I found something to think about in the last part of discussion (line 221-228). It could be useful to anticipate this concept in the first part of discussion. 

Please check these lines:

55-56 and 95-96: too many repetition of the word "such" in these lines; maybe you could use some different words.

113: correct the word "metabolomic"; miss the letter "o". 

117: correct unit of measurement after "400". 

Finally, can you add a PET/CT image in this article? It would be nice, I think. 

Good work. 

Author Response

We thank all fruitful comments to improve our manuscript. We heighted the revised sentences in red.

We asked the professional English proof editor to revise our manuscript throughout whereas this revisions were not colored.

Reviewer’s comments

This study rises the issue of screening (in patients with risk factor in oral cancer) before the execution of PET/CT. This could be very interesting because of the lack of specificity in FDG PET (tumour vs inflammation). For this reason I found something to think about in the last part of discussion (line 221-228). It could be useful to anticipate this concept in the first part of discussion.

Our comments

Thank you for your helpful comments. According to the reviewer’s comments, we have revised the first and second paragraphs of the discussion to add further details on the concept.

Page 8, lines 14–19.

This study aimed to determine the relationship between salivary metabolites and SUVmax of 18F-FDG PET/CT. 18F-FDG PET/CT is a cancer screening tool, but it has several limitations. Thus, there is a need for another screening tool that is complementary to 18F-FDG PET/CT. We have evaluated the screening potential of saliva samples and the correlations between SUVmax and salivary metabolites in patients with oral cancer.

Page 9, lines 1–2.

The salivary metabolites observed in this study may be used as a first-line screening tool to confirm the presence of oral cancer before PET/CT.

55-56 and 95-96: too many repetition of the word "such" in these lines; maybe you could use some different words.

Our comments

Thank you for your helpful comments.

According to your suggestions, we have used different words instead of “such” wherever possible at the following instances:

Page 3, lines 8–11.

Hence, if salivary metabolites demonstrate a significant correlation with PET parameters such as the maximum standard uptake value SUVmax, they may have the potential to be used as a screening tool prior to PET/CT to identify patients with high SUVmax.

Page 4, line 23 to page 5, line 4.

Therefore, we believe that it is clinically meaningful to analyze the relationship between salivary metabolites and PET parameters such as SUVmax in patients with oral cancer. If salivary metabolites demonstrate a significant correlation with PET parameters such as SUVmax, they may have the potential to be used as a screening tool prior to PET/CT to identify patients with high SUVmax. i.e. the presence of oral cancer.

113: correct the word "metabolomic"; miss the letter "o".

Our comments

Thank you for your helpful comments. We have corrected this spelling mistake.

117: correct unit of measurement after "400".

Our comments

Thank you for your helpful comments. We have revised the unit to “micro L.”

Finally, can you add a PET/CT image in this article? It would be nice, I think.

Our comments

We agree with your suggestions. However, the participants were recruited from the Department of Dentistry, Oral and Maxillofacial Plastic and Reconstructive Surgery of Yamagata University Hospital from 2012 to 2014. We have to obtain informed consent again for using PET/CT images of patients with oral cancer. This is difficult now. I appreciate your understanding.

Good work.

Thank you for your encouraging comments. We appreciate your kind and helpful review.

Reviewer 2 Report

The manuscript submitted by Ishikawa et al. reports the analysis of correlation between salivary metabolites and SUVmax of 18F-FDG from oral cancer patients. They discovered that higher salivary metabolite concentrations are positively correlated to higher SUVmax of 18F-FDG, and concluded that analysis of salivary metabolites could be used to select oral cancer patients for further 18F-FDG PET images. This study is novel and the manuscript is well written. However, there are a number of deficiencies as listed below:

Major deficiencies:

  • Sex should be a variable in the analysis, and the results should be presented and discussed.
  • Tumor stage should be a variable in the analysis, and the results should be presented and discussed.
  • Tumor volume should be a variable in the analysis and the results should be presented and discussed.
  • As the same lab has previously reported the correlation of salivary metabolites and the presence of oral cancer, it is unclear the extra value of this study on the diagnosis of oral cancer.

Minor deficiencies:

  • Change all “standard uptake value” to “standardized uptake value”.
  • Line 48: The 18 in “18F-FDG” should be superscript.
  • Line 113: “The metablomics data ...” should be “The metabolomics data ...”.
  • Line 117: “On average, 400 l of unstimulated whole saliva was collected”. Please double check the volume, it is unlikely to collect 400 l of saliva from a human subject.
  • Line 154: unbold “Subjects’ characteristics are shown in Table 1.”
  • Table 1: change “average(SD)” to “average”.
  • Table 1: Is the “Standard Uptake Value” in Table 1 “Average SUVmax”?
  • Line 165: delete “**p = 0.008”.

Author Response

We thank all fruitful comments to improve our manuscript. We heighted the revised sentences in red.

We asked the professional English proof editor to revise our manuscript throughout whereas this revisions were not colored.

The manuscript submitted by Ishikawa et al. reports the analysis of correlation between salivary metabolites and SUVmax of 18F-FDG from oral cancer patients. They discovered that higher salivary metabolite concentrations are positively correlated to higher SUVmax of 18F-FDG, and concluded that analysis of salivary metabolites could be used to select oral cancer patients for further 18F-FDG PET images. This study is novel and the manuscript is well written. However, there are a number of deficiencies as listed below:

Major deficiencies:

Sex should be a variable in the analysis, and the results should be presented and discussed. Tumor stage should be a variable in the analysis, and the results should be presented and discussed. Tumor volume should be a variable in the analysis and the results should be presented and discussed.

According to the reviewer’s comments, we analyzed and discussed the discrimination ability of each metabolite with these variables. Sex was obtained in both patients and controls, and therefore, we developed another MLR model including metabolites and sex and evaluated the effect of sex. The discrimination ability of the MLR model was evaluated for the early stage and the advanced stage of the patients of oral cancers. To describe these analyses, Figure 1 was modified and panels b and c were added. The result was revised (p8, lines 1-11). The discussion and the legends for Fig. 1 were also revised (p9, lines 9-14, and p13, lines 13-19).

As the same lab has previously reported the correlation of salivary metabolites and the presence of oral cancer, it is unclear the extra value of this study on the diagnosis of oral cancer.

We thank these comments. To clarify the difference between the previous and current studies, we revised the last paragraph of the discussion. The defined the reason why we used a different approach to identify the discrimination markers. We also expand the limitation and cited a new reference 32. (p11, lines 13-20 and p17, lines 3-4)

Minor deficiencies:

Change all “standard uptake value” to “standardized uptake value”.

According to the reviewer’s comment, we revised this word throughout the manuscript.

Line 48: The 18 in “18F-FDG” should be superscript.

We thank this observation. We revised this word throughout the manuscript.

Line 113: “The metablomics data ...” should be “The metabolomics data ...”.

We thank this observation. We fixed this typo.

Line 117: “On average, 400 l of unstimulated whole saliva was collected”. Please double check the volume, it is unlikely to collect 400 l of saliva from a human subject.

We thank this observation. We revised the unit to micro L. (page 6, Line 2)

Line 154: unbold “Subjects’ characteristics are shown in Table 1.”

We thank this observation. We revised this word. (page 6, Line 2)

Table 1: change “average(SD)” to “average”.

We thank this observation. We revised this word at Table 1.

Table 1: Is the “Standard Uptake Value” in Table 1 “Average SUVmax”?

We thank this observation. We exchanged this word to Average SUVmax at Table 1.

Line 165: delete “**p = 0.008”.

We thank this observation. We added a new column to describe the p-value to the Table and also revise this word to **P < 0.01 at the end of the Note sentence.

Round 2

Reviewer 2 Report

This is a revised version, and the authors have provided satisfactory responses and made changes based on the Reviewer's comments. Therefore, this revised version could be accepted for publication without further changes.